# High Doses of Pesticides Induce mtDNA Damage in Intact Mitochondria of Potato In Vitro and Do Not Impact on mtDNA Integrity of Mitochondria of Shoots and Tubers under In Vivo Exposure

**DOI:** 10.3390/ijms23062970

**Published:** 2022-03-10

**Authors:** Alina A. Alimova, Vadim V. Sitnikov, Daniil I. Pogorelov, Olga N. Boyko, Inna Y. Vitkalova, Artem P. Gureev, Vasily N. Popov

**Affiliations:** 1Department of Genetics, Cytology and Bioengineering, Voronezh State University, 394018 Voronezh, Russia; aa10022607@gmail.com (A.A.A.); vvs-96@yandex.ru (V.V.S.); pogorelov.d12@gmail.com (D.I.P.); olga.boiko2000@yandex.ru (O.N.B.); vitkalovai@inbox.ru (I.Y.V.); pvn@bio.vsu.ru (V.N.P.); 2Laboratory of Metagenomics and Food Biotechnology, Voronezh State University of Engineering Technologies, 394036 Voronezh, Russia

**Keywords:** imidacloprid, metribuzin, *Solanum tuberosum*, mitochondrial DNA, field experiment, hydrogen peroxide, aconitate hydratase, succinate dehydrogenase

## Abstract

It is well known that pesticides are toxic for mitochondria of animals. The effect of pesticides on plant mitochondria has not been widely studied. The goal of this research is to study the impact of metribuzin and imidacloprid on the amount of damage in the mtDNA of potato (*Solanum tuberosum* L.) in various conditions. We developed a set of primers to estimate mtDNA damage for the fragments in three chromosomes of potato mitogenome. We showed that both metribuzin and imidacloprid considerably damage mtDNA in vitro. Imidacloprid reduces the rate of seed germination, but does not impact the rate of the growth and number of mtDNA damage in the potato shoots. Field experiments show that pesticide exposure does not induce change in aconitate hydratase activity, and can cause a decrease in the rate of H_2_O_2_ production. We can assume that the mechanism of pesticide-induced mtDNA damage in vitro is not associated with H_2_O_2_ production, and pesticides as electrophilic substances directly interact with mtDNA. The effect of pesticides on the integrity of mtDNA in green parts of plants and in crop tubers is insignificant. In general, plant mtDNA is resistant to pesticide exposure in vivo, probably due to the presence of non-coupled respiratory systems in plant mitochondria.

## 1. Introduction

Archeological and genetic research indicates that potato domestication (*Solanum tuberosum* L.) started about 8000 years ago on the border of present-day Peru and Bolivia [1]. According to FAOSTAT to 2019, the largest quantities of potatoes are produced in China (91.8 million tons), India (50.2 million tons), Russia (22.1 million tons), Ukraine (20.3 million tons), and the USA (19.2 million tons) [2]. It is noteworthy that from 1961 to 2019 the area of potato fields decreased by 21.7% (from 22,147,976 ha to 17,340,986 ha), at the same time world production of potato increased by 36.9% (from 270,552,196 tons to 370,436,581 tons). So, 12.2 tons of potatoes were harvested from one hectare of field in 1961, while 21.4 tons of potatoes were harvested from one hectare of field in 2019 [2]. The widespread use of pesticides was the main reason for the increase in potato yields [3].

However, significant progress in yield protection was halted by the evolution of resistance against pesticides in pest populations. Synthetic and natural pesticides have thus become crucial; however, resources for pest control are non-renewable. The development of resistance led to the development and wider use of more and more pesticides [4]. In turn, the even wider use of pesticides led to environmental consequences. The excessive use of pesticides may lead to the destruction of biodiversity [5]. Pesticides can be toxic, not only for pests and plant-disease fungi, but also for other non-target organisms, including birds [6], fish [7,8], insect pollinators [9,10], and non-target plants [11,12]. Moreover, pesticides affect human health as a result of pollution of the environment and food. The effects of pesticides on humans include headaches and nausea; skin and eye irritation [13]; and chronic problems, such as cancer [14], neurodegenerative diseases, and, especially, Parkinson’s disease [15]. 

Mitochondria are among the main targets for pesticides. The majority of the most commonly used insecticides and fungicides inhibit the complex I [16], while less inhibit complex III [17], complex IV [18], and ATPase [19], and only several pesticides inhibit complex II [20] of electron transport chain (ETC). Pesticides may act as an uncoupler of oxidative phosphorylation [21], inducers of apoptosis, which are closely related with mitochondrial dysfunction [22]. Most herbicides inhibit electron flow within the photosynthetic ETC, in particular photosystem II [23], but can cause mitochondrial dysfunction too [24]. The effect of insecticides, fungicides, and herbicides on the induction of oxidative stress is well established [9,24,25]. Pesticide-induced oxidative stress causes damage to some mitochondrial components, in particular mitochondrial DNA (mtDNA) [9,10,26].

Despite significant progress in the research of pesticide effect on animal mitochondria and mtDNA, there is a gap in the knowledge of the effect of pesticides on plant mitochondria. Meanwhile, the mitochondria of plants and animals differ significantly, both in the features of the functioning of the electron transfer chain [27] and in the structure of mtDNA [28]. For most eukaryotic organisms, including plants, the mtDNA is a single circular molecule [29,30]. Recently, it was found that potato mitogenome has a multi-partite architecture, divided in at least three independent molecules [30]. The length of the first chromosome is 312,491 bp. It encodes 54 genes, including 13 subunits of the I complex of the ETC (*nad1e*, *nad2cde*, *nad4*, *nad6*, *nad7*, *nad5ab*, *nad4L*, *nad1d*, *nad5de*, *nad1a*, *nad5c*, *nad1bc*, *nad1d*); 2 subunits of the IV complex (*cox2*, *cCC*); 4 subunits of the ATPase (*atp1*, *atp4*, *atp6*, *atp9*); 6 transmembrane proteins (*mttb*, *rps4*, *rps19*, *rps3*, *rps13*, *rpl16)*; 5S, 18S, and 26S ribosomes (*5S*, *18S*, *26S*); 1 maturase (*matR*); and 18 open reading frames (ORFs*)*. The second chromosome has a length of 112,797 bp. It encodes 15 genes, including 1 subunit of the ETC complex I (*nad3*), 1 subunit of the II complex (*sdh4*), 3 subunits of the III complex (*ccmFN*, *ccmFC*, *cob*), 2 subunits of the complex IV (*cox1*, *cox3*), 1 subunit of the ATPase (*atp8*), 5 transmembrane proteins (*rps1*, *rps10*, *rps14*, *rps12*, *rpl5*), as well as 2 ORFs. The third chromosome has a length of 49,229 bp. It encodes seven genes, including two subunits of the ETC complex I (*nad2ab*, *nad9*), one subunit of complex II (*sdh3*), one subunit of complex III (*ccmB*), five transmembrane proteins (*rpl10*, *rpl2*), as well as one ORF [30].

Our previous screening research showed that pesticides have a high potential for toxicity to potato mitochondria in vitro [31]. However, there are practically no data on the toxicity of pesticides on plant mitochondria in vivo. For the study, we used two pesticides. Imidacloprid was first registered for use as an insecticide in the USA in 1994 and it is currently one of the best-selling insecticides [32]. In Russia alone, at least 84 commercial pesticides based on imidacloprid are sold [33]. Metribuzin is an effective organic herbicide utilized to inhibit the growth of the broadleaf weed and dangerous grasses. Metribuzin is a widely used pesticide around the world and in many countries has replaced many types of common organic herbicides such as chlorinated hydrocarbons [34]. In Russia, metribuzin is available as 15 commercial pesticides [35]. Our previous research showed that 200 µM of imidacloprid and metribuzin induced mtDNA damage [31], but regions or chromosomes, which are most susceptible to lesions, are currently unidentified. We hypothesized that three mtDNA chromosomes will differ susceptible to damage. At the same time, the effect of pesticides in vitro and in vivo can differ significantly, since plant mitochondria have a number of features which may mediate resistance to pesticides. Recently, we showed that alternative NADH dehydrogenases is a factor responsible for plant resistance to xenobiotics, such as mitochondria-targeted pesticides. The permeabilized mitochondria are more sensitive for pesticide exposure compared to intact mitochondria [36]. We can assume that in vivo plant mitochondria may be significantly more resistant to intact plant mitochondria and animal mitochondria under pesticides exposure.

The goal of this research is to develop a method for identifying mtDNA regions that are most susceptible to lesions induced by exogenous compounds; studying the effect of imidacloprid and metribuzin on mtDNA damage in vitro and in vivo; and evaluating this pesticide effect on the potato germination, shoot growth, rate of reactive oxygen species (ROS) production, and the activity of mitochondrial enzymes in the potato tubers. The results obtained will make it possible to judge about plant mitochondria susceptible to pesticide-induced damage in vitro compared to in vivo condition.

## 2. Results

### 2.1. Optimization of the Method for Estimating of the mtDNA Damage Amount

Isolated potato mitochondria were incubated with 500 μM of H_2_O_2_ for 30 min to induce lesions in mtDNA. H_2_O_2_ induced lesions to mtDNA in situ, encoding the *26s* rRNA gene (F (1,18) = 44.115, *p* = 0.0001) of the first mtDNA chromosome. The median number of lesions per 10 kb of mtDNA was 4.45 (4.12; 4.95), *p* = 0.005. Furthermore, statistically significant lesions were recorded for the *cox1* gene fragment (F (1,18) = 20.3582, *p* = 0.0003) of the second chromosome. The median amount of damage per 10 kb of mtDNA was 6.62 (4.98; 6.78) (*p* = 0.00015). For the third chromosome of mtDNA, H_2_O_2_-induced lesions were observed in the *sdh3* gene fragment (F (1,18) = 25.2873, *p* = 0.0003), and the median amount of damage per 10 kb of mtDNA was 5.42 (4.56; 6.27), *p* = 0.0003. The H_2_O_2_-induced increase in the amount of mtDNA damage for genes *cox2* (F (1,18) = 23.5366, *p* = 0.0548) and *cob* (F (1,18) = 31.7197, *p* = 0.062) was statistically insignificant. There was no statistically significant change in the number of mtDNA lesions for the region, which encoded *ccmB* (F (1,18) = 1.8968, *p* = 0.578) (Figure 1).

We did not find a correlation between the number of H_2_O_2_-induced mtDNA of potato and frequency of occurrence of GTGR (r_s_ = −0.6) or RTGR (r_s_ = −0.31) sequences (where R is A or G). It should be noted that H_2_O_2_-induced mtDNA was statistically significant (*p* < 0.05) or close to statistically significant (the *p* value equals somewhere between 0.05 and 0.06) in all studied mtDNA fragments, except for *ccmB* (*p* = 0.58) (Figure 1). The amount of mtDNA damage induced by H_2_O_2_ is characterized by an extremely high coefficient of variation (CV) (CV = 177 for *ccmB*, while CV ranged from 14 to 46 in other fragments).

### 2.2. Pesticides Effect on the Amount of mtDNA Damage In Vitro

Both metribuzin and imidacloprid damaged the potato mitochondrial genome in vitro for all studied mtDNA regions (Figure 2A). Typically, more lesions were induced by the metribuzin. For the region encoding the *cob* gene (F (2,17) = 6.1623, *p* = 0.0097), more mtDNA lesions were induced by the imidacloprid (the median amount of damage per 10 kb of mtDNA was 4.39 (4.15; 4.58), *p* = 0.002). Metribuzin-induced damage of the *cob* gene-encoded fragment was not statistically significant (the median amount of damage per 10 kb of mtDNA was 4.01 (2.65; 4.28), *p* = 0.12 compared with control) (Table 1).

The same number of lesions was induced by pesticides in the *cox2* gene (F (2,17) = 10.1543, *p* = 0.0013). The median amount of damage per 10 kb of mtDNA was 4.58 (3.58; 4.77) for the metribuzin-treated mitochondria (*p* = 0.013). The median amount of damage per 10 kb of mtDNA was 4.56 (4.14; 4.70), for the imidacloprid-treated mitochondria (*p* = 0.016). Statistically significant differences were recorded for the *26s* rRNA gene (F (2,17) = 20.6552, *p* = 0.00003). The median number of metribuzin-induced damage per 10 kb of mtDNA was 4.35 (3.56; 4.48), *p* = 0.002, and the median number of imidacloprid-induced damage per 10 kb of mtDNA was 4.26 (3.95; 4.59), *p* = 0.002 (Table 1).

There were changes in the amount of damage in the region encoding the *cox1* gene (F (2,17) = 39.88, *p* = 0.00001). The median number of metribuzin-induced damage per 10 kb of mtDNA was 4.8 (4.28; 5.03), *p* = 0.0006, and the median number of imidacloprid-induced damage per 10 kb of mtDNA was 4.55 (4.03; 4.78), *p* = 0.002. A similar trend was observed for the *ccmB* and *sdh3* genes located on the third chromosome. For *ccmB* (F (2,17) = 17.0728, *p* = 0.00009), the median number of metribuzin-induced damage per 10 kb of mtDNA was 4.72 (3.89; 5.23), *p* = 0.001, and the median number of imidacloprid-induced damage per 10 kb of mtDNA was 4.22 (3.63; 4.38), *p* = 0.005. For *sdh3* (F (2,17) = 6.5538, *p* = 0.0078), the median number of metribuzin-induced damage per 10 kb of mtDNA was 4.96 (3.84; 5.10), *p* = 0.015, and the the median number of imidacloprid-induced damage per 10 kb of mtDNA was 4.24 (3.80; 4.51), *p* = 0.005 (Table 1).

### 2.3. Effect of Pesticides on Seed Germination and Potato Growth under Controlled Conditions

We found that the control (1 and 2) and imidacloprid-treated (3 and 4) groups had the same technical germination—82.1%. Germination was recorded only on the 16th day in the imidacloprid-treated group, which indicated a delay in comparison with the plants of control group, which germinated even on the 2nd day. The start of germination curves in both groups is spasmodic. Then, the curves of germination reach a plateau. The germination maximum was equal after 35 days (Figure 3A). The germination energy in the control group was 71.4%, in the imidacloprid-treated group—78.6%.

Imidacloprid treatment affected seed germination. So, in the control group, 50% of the tubers sprouted on the fourth day. In the group treated with imidacloprid, 50% of the tubers sprouted on the 20th day (Figure 3A). Differences in the size of potato seedlings were observed in the first weeks of the experiment, in particular, on the 20th day of the experiment (F (1,54) = 8.2663, *p* = 0.00577), when the median of the control seedlings was 0.5 cm (0.15; 0.9), and the average length of potato seedlings treated with imidacloprid was 0 cm (0; 0.4) (Figure 3B). However, on the 38th day of the experiment (the day of metribuzin treatment of some experimental groups), no differences in the growth of seedlings were observed (F (1,54) = 1.5644, *p* = 0.21642) (Figure 3C). Metribuzin treatment also did not affect the growth dynamics of potato seedlings. In particular, there were no differences in the length of seedlings on the 52nd day of the experiment (F (3,52) = 0.04674, *p* = 0.98645) (Figure 3D). On the 52nd day of the experiment, there were no statistically significant differences in the mass of the root system of the seedlings (F (3,52) = 1.9549, *p* = 0.1322) (Figure 4).

### 2.4. Effect of Pesticides on the Amount of mtDNA Damage in Potato Shoots

There was a statistically significant difference in the mean number of mtDNA damage in the potato shoots (F (3,652) = 7.6758, *p* = 0.00005). Tuckey’s post-hoc test showed a statistically significant decrease in the number of lesions per 10 kb of mtDNA of shoots treated with imidacloprid (median 2.16 (2.33; 4.0)), compared with shoots treated with metribuzin (median 3.53 (1.06; 4.72), *p* = 0.038) (Figure 2B).

For the mtDNA region encoding the *26s rRNA* gene, a pesticide-induced decrease in the amount of mtDNA lesions was observed (F (3,120) = 8.38, *p* = 0.00004). There was a significant decrease in the amount of damage per 10 kb of mtDNA in metribuzin (median 1.64 (0.11; 2.71), *p* = 0.0008), imidacloprid (median 1.21 (−1.31; 3.22), *p* = 0.0014), and imidacloprid–metribuzin (median 1.96 (−0.88; 2.99), *p* = 0.0002) groups compared with the control group (median 4.3 (2.04; 4.81)). No significant differences were founded for mtDNA regions encoding genes *cox2* (F (3,120) = 1.108, *p* = 0.35) and *cox1* (F (3,108) = 1.232, *p* = 0.3). There was a statistically significant (F (3,116) = 3.99, *p* = 0.0095) decrease in the number of mtDNA lesions per 10 kb of mtDNA in the *cob* gene in the imidacloprid-treated group (median 1.53 (−0.99; 3.59)) compared with the metribuzin-treated group (4.43 (3.29; 4.73), *p* = 0.006). There was a statistically significant (F (3,104) = 7.2946, *p* = 0.0002) decrease in the number of mtDNA lesions per 10 kb of mtDNA in the site encoding the *ccmB* gene in the imidacloprid group (median −5.11 (−5.23; 3.95)) in comparison with all other groups—i.e., control (median 3.45 (−1.22; 4.49), *p* = 0.0067), metribuzin (median 4.2 (1.43; 4.79), *p* = 0.00067), and imidacloprid–metribuzin (median 4.43 (1.09; 4.83), *p* = 0.00021) groups. There were significant differences in the region encoding the *sdh3* gene (F (3,108) = 3.2145, *p* = 0.026). There was a decrease in the number of mtDNA lesions per 10 kb of mtDNA for the metribuzin group (median 4.76 (4.60; 5.10)) compared with the control (median 4.80 (2.48; 4.97), *p* = 0.04), and an increase compared with the imidacloprid–metribuzin group (median 4.65 (0.84; 4.89), *p* = 0.04) (Table 2).

### 2.5. Effect of Pesticides on the Potato Growth and the Weight of the Crop in the Field Experiment

The median weight of the crop in the control group was 95 g (47.5; 149.5), as well as in the metribuzin group (median 100 g (69; 164)). There was a tendency to increase the crop weight in the group of potatoes, which was treated with imidacloprid. The median weight of crop was 222 g in the imidacloprid+/− group (42; 285), 120 g in the imidacloprid−/+ (109; 187.5), and 133.5 g in the imidacloprid+/+ (98.5; 229). However, there was no statistically significant difference (F (4, 31) = 0.9549, *p* = 0.4459) (Figure 5B).

Unlike potato weight, we did not show any tendency in the spouts length. The median length was 27 cm in the control group (15; 37), 28 cm in the imidacloprid+/− group (21; 36), 26.5 cm in the imidacloprid−/+ group (15; 32), 26 cm in the imidacloprid+/+ group (14, 35), 25 cm in the metribuzin group (15; 30). There was no significant difference (F (436) = 0.6236, *p* = 0.6486) (Figure 5A).

### 2.6. Effect of Pesticides on the Amount of mtDNA Damage in Potato Tubers

Pesticide treatment can impact the mean amount of the mtDNA damage in potato tubers (F (4,723) = 16.9697, *p* = 0.00001). A statistically significant increase in the number of mtDNA lesions per 10 kb of mtDNA was found in the imidacloprid−/+ group (median 4.25 (2.75; 4.78), *p* = 0.0041) and the imidacloprid+/− group (median 3.92 (1.54; 4.77), *p* = 0.048) compared with the control (median 4.18 (4.72; 4.80)). There was a decrease in the number of mtDNA lesions per 10 kb of mtDNA in the imidacloprid+/+ group (median 1.31 (−3.24; 4.68)), in comparison with the imidacloprid−/+ group (*p* = 0.0013) (Figure 2C).

For the region encoding the *26s* rRNA gene, statistically significant differences in the number of mtDNA lesions were noted (F (4,123) = 6.22, *p* = 0.0001). The number of lesions per 10 kb of mtDNA increased in the imidacloprid−/+ group (median 4.16 (3.68; 4.45)) compared with the control (median 3.26 (−4.89; 5.00), *p* = 0.0019), and decreased in the imidacloprid+/− group (median 1.54 (−0.92; 3.00)) and imidacloprid+/+ (median 1.31 (−1.17; 3.18)) compared with the imidacloprid−/+ group (*p* = 0.00048 and *p* = 0.004, respectively). No statistically significant changes in the amount of mtDNA lesions were found for the *cox2* gene (F (4,123) = 2.37) (Table 3).

Statistically significant differences were found in the number of lesions in the mtDNA region encoding the *cob* gene (F (4,91) = 4.58, *p* = 0.0021). The median number of lesions per 10 kb of mtDNA was 4.73 (4.68; 4.74) in the imidacloprid+/+ group, which is statistically significantly higher than in the metribuzin group (median 4.70 (−4.70; 4.73), *p* = 0.019). Differences in the number of lesions were shown for the mtDNA fragment encoding the *cox1* gene (F (4,115) = 8.77, *p* = 0.00001). A significant increase in the number of mtDNA lesions per 10 kb of mtDNA was shown in the imidacloprid−/+ group (median 4.93 (3.73; 5.01), *p* = 0.011) and the imidacloprid+/− group (median 4.90 (4.50; 5.00), *p* = 0.002) compared with the control group (median 1.69 (−4.78; 4.31)), as well as compared with the metribuzin-treated group (2.77 (−5.03; 4.90), *p* = 0.04 and *p* = 0.01, respectively) and the imidacloprid+/+ group (−0.33 (−3.20; 2.52), *p* = 0.001 and *p* = 0.0002, respectively) (Table 3).

There were differences in the amount of mtDNA lesions in the *ccmB* gene (F (4,123) = 7.10, *p* = 0.00004). The median number of mtDNA lesions per 10 kb of mtDNA in the imidacloprid+/+ group was 0.00 (−4.78; 4.45), which was less than in the metribuzin group (median 4.60 (3.15; 5.07), *p* = 0.00038), the imidacloprid−/+ group (median 4.82 (3.44; 5.10), *p* = 0.000045), and the imidacloprid+/− group (median 4.71 (1.72; 5.13), *p* = 0.0007). Similar changes in the amount of mtDNA lesions were also shown for the mtDNA fragment encoding the *sdh3* gene (F (4,23) = 11.89, *p* = 0.00001). In the imidacloprid+/+ group (median −3.84 (−5.07; 1.99)), the number of lesions per 10 kb of mtDNA was lower compared to the control (median 4.35 (−4.45; 4.93), *p* = 0.052), as well as experimental groups, i.e., metribuzin (median 4.05 (3.40; 4.40), *p* = 0.000017), imidacloprid−/+ (median 3.77 (2.25; 4.56), *p* = 0.000017), and imidacloprid+/− (median 4.71 (1.72; 5.13), *p* = 0.000018) groups (Table 3).

### 2.7. Effect of Pesticides on Enzyme Activity in Potato Tubers

Statistically significant differences in activity values were found for succinate dehydrogenase (F (4,26) = 2.6392; *p* = 0.0566) (Figure 6A). The enzyme activity in the imidacloprid−/+ group increased (median 0.00051 mmol/min/g of FW (0.000265; 0.000794)) compared to the imidacloprid+/− group (median 0.00178 mmol/min/g of FW (0.001305; 0.002381); *p* = 0.02512). No statistically significant changes were found for aconitate hydratase (F (4,26) = 0.5365; *p* = 0.7101) (Figure 6B).

### 2.8. Effect of Pesticides on the Rate of H_2_O_2_ Production in Potato Tubers

Analysis of the data obtained during the assessment of the rate of H_2_O_2_ production without oligomycin showed statistically significant differences between groups (F (4,39) = 38.0792; *p* = 0.00001) (Figure 7A). Mitochondria of the control group had maximal values of the rate of the H_2_O_2_ production (median 181 pmol/min/mg of protein (149; 196)). The minimal rate of the H_2_O_2_ production was in mitochondria from imidacloprid+/+ tubers (median 49 pmol/min/mg of protein (44; 55), which was significantly less compared with the control group (*p* = 0.000126). The metribuzin and imidacloprid−/+ groups also had a lower rate of H_2_O_2_ production compared with the control group (median 121 pmol/min/mg of protein (104; 129) (*p* = 0.000136) and 135 pmol/min/mg of protein (129; 145) (*p* = 0.002271), respectively). The rate of the H_2_O_2_ production in the imidacloprid+/− group was almost equal to the control group (median pmol/min/mg of protein 174 (158; 193) (*p*= 0.966632)) (Figure 7A).

Similar results were obtained for mitochondria, which were inhibited by oligomycin (F (4,39) = 47.1607; *p* = 0.00001) (Figure 7B). The maximal rate of the H_2_O_2_ production was observed for the control (median 171 pmol/min/mg of protein (159; 186)) and imidacloprid+/− (median 180 pmol/min/mg of protein (165; 194)) groups. For other groups, we found a statistically significant decrease in the rate of the H_2_O_2_ production (median for metribuzin group was pmol/min/mg of protein (104; 123), *p* = 0.000133. The median for imidacloprid−/+ was 140 pmol/min/mg of protein (131; 153), *p* = 0.006312, and the median for imidacloprid+/+ was 62 pmol/min/mg of protein (53; 71), *p* = 0.000126) (Figure 7B).

## 3. Discussion

### 3.1. Identification of mtDNA Regions That Are More Prone to Oxidative Damage

It is well known that mtDNA is more prone to oxidative damage than nuclear DNA [37]. The proximity of mtDNA to the sites of oxidative phosphorylation and ROS production and the lack of the protection by histones are the main reasons why mtDNA is susceptible to damage [38]. For this reason, mtDNA damage is a popular marker of oxidative stress and an indicator of the level of degenerative processes during pathogenesis of various diseases in animal models [39,40,41] and humans [42]. There are data which show that pesticides can induce mtDNA damage in insects [10], mammals [43], and human’s cell cultures [16]. At the same time, the ability of plant mtDNA to accumulate oxidative damage is practically not studied in comparison with animals. 

There are a number of methods to estimate DNA damage. The classical methods, such as Southern blot and high-performance liquid chromatography, have a number of limitations. Long-range PCR has a number of advantages in measuring mtDNA damage, including its high sensitivity and ability to measure mtDNA damage in specific regions based on primer design [44]. We have previously shown that H_2_O_2_ induced significant damage to mtDNA; however, the distribution of damage is unequal along the molecule [45]. D-loop, the non-coding regulatory region of animal’s mtDNA, is the most susceptibility region for somatic mutations [46] and oxidative damage [45,47]. Plants most likely employ multiple mechanisms to replicate the mtDNA due to the complex structure of the mitochondrial genome; nevertheless, such a structure in the D-loop appears to be missing [48]. Among the three studied potato chromosomes, we did not identify a region that would be most susceptible to H_2_O_2_-induced damage relative to other regions (Figure 1). It was previously shown that regions of DNA that are enriched in RTGR (where R is A or G) sequences are more sensitive to H_2_O_2_-induced oxidative damage due to the specific interactions of Fe^2+^ ions with thymine in the purine–T–G–purine context which make the RTGR sequence more prone to the Fenton reaction [49,50]. Furthermore, the correlation between ROS-induced DNA damage and the frequency of occurrence of GTGR (i.e., GTGG and GTGA) has been shown [51]. Then, confirmed these data was confirmed in mtDNA of mice and has been shown strong correlations between the H_2_O_2_-induced number of mtDNA damage and the frequency of occurrence of GTGR sequences in the mtDNA of brain (rs = 0.78) and liver (rs = 0.9) [45].In this research, we did not find a correlation between the number of H_2_O_2_-induced mtDNA damage of potato and the frequency of occurrence of GTGR (r_s_ = −0.6) or RTGR (r_s_ = −0.31) sequences. It should be noted that H_2_O_2_ induced significant damage in all mtDNA fragments, except for *ccmB* (Figure 1). In this fragment, the amount of mtDNA damage induced by H_2_O_2_ is characterized by an extremely high coefficient of variation compared with the other fragments. The *ccmB* gene in potato mtDNA is characterized by the fact that about 50% of *ccmB* transcripts cannot be translated into functional proteins [30]. It is likely that this region of mtDNA contains a component that affects the efficiency of not only RNA polymerase, but also DNA polymerase. For this reason, a high variation coefficient of the amount of mtDNA damage is observed when studying this mtDNA fragment.

### 3.2. Effect of Metribuzin and Imidacloprid on Potato mtDNA In Vitro

When evaluating the results of an in vitro experiment, it was found that the damage to mtDNA obtained after treatment with metribuzin and imidacloprid at a concentration of 200 μM is equivalent to the damage induced by treatment with H_2_O_2_ at a concentration of 500 μM (Figure 2A, Table 1). We have previously shown similar results when treating mouse brain mitochondria with 500 μM H_2_O_2_ and 1 μM rotenone pesticide, where rotenone caused even slightly more damage than H_2_O_2_ [52]. It is believed that mtDNA damage is mainly initiated by ROS [53]. It is known that rotenone, which inhibits ETC complex I of animal mitochondria, causes overproduction of H_2_O_2_ [54]. However, we have previously shown that imidacloprid and metribuzin did not increase the rate of ROS production in vitro [31], but they initiate serious damage. It is likely that the mechanism of DNA-related toxicity of pesticides is not only associated with the induction of oxidative stress. It is known that most pesticides are electrophilic substances, which ensures their high reactivity towards DNA. Pesticides induced adduct formation due to the binding with DNA. Reactions of DNA bases with pesticides or their metabolites alter its structure and prevent DNA replication [55]. So, we can assume that the mechanism of pesticide toxicity in vitro is not associated with ROS production, and it is most likely that pesticides directly interact with mtDNA and damage it.

### 3.3. Effect of Metribuzin and Imidacloprid on Potato mtDNA In Vivo

However, we have shown that pesticide treatment does not cause damage to mtDNA in the green parts of plant seedlings (Figure 2B). In contrast, a pesticide-induced reduction in the number of lesions was observed in the mtDNA region encoding the *26s* RNA gene. Imidacloprid treatment caused a decrease in the amount of mtDNA damage in the mtDNA region encoding the *ccmB* gene. However, the imidacloprid has an effect on the germination rate of potato seeds. In the control group, 50% of the tubers germinated on day 4, while in the imidacloprid-treated group they did so on day 20 (Figure 3A). In the first days of the experiment, the growth rate of seedlings in the imidacloprid-treated group was lower than in the control; however, by the middle of the experiment, the experimental seedlings “caught up” to control seedlings. Previously it has been shown that the average imidacloprid-induced decline in normal germination of rice seed was 18.4% in 12 of the 15 cultivars or rice. However, there were generally no significant differences in shoot lengths and root system dry weights between control and pesticide-treated plants [56].These data are consistent with our results. We did not find statistically significant differences in shoot lengths on the 38th day of the experiment and root weight at the end of the experiment (Figure 3C and Figure 4). It can be assumed that the treatment of seeds with an imidacloprid causes some decline, which affects the rate of germination. However, in green seedlings, these effects are leveled due to the presence of non-coupled respiration systems in plant mitochondria, primarily alternative oxidases (AOX). AOX can probably participate in the detoxification of pesticides by reducing the level of oxidative stress [57]. It is well known that AOX acts as a metabolic and signaling regulator and can maintain homeostasis, which is most important under stress conditions [58]. Pesticide exposure can be considered as severe abiotic stress for plants [59]. Nevertheless, the role of AOX in the regulation of plant response on pesticide toxicity needs further study.

However, there was research which showed the significant impact of the pesticide on potato germination and growth. Bocharov and Fomin showed that insecticides Kruiser (thiamethoxam) and Prestige (imidacloprid, pencycuron), in a mixture with the fungicide Maxim (fludioxonil), inhibited growth processes at the initial stage of ontogenesis (germination-seedlings) [60]. Pesticides reduced the length of regeneration seedlings and their roots (total and working surface of the root system) [60]. The high concentration (2–4 times higher than recommended dose) of some commonly used pesticides (including imidacloprid) decreased seed germination, biomass production, and photosynthetic pigments in tomato. However, at lower doses (quarter and half of the recommended dose), the pesticides had some stimulatory effects on growth as compared to the control [61]. These data do not fundamentally contradict our results, which showed that high doses of imidacloprid (three times higher than the recommended dose) negatively affect seed germination (Figure 3A); however, we have not shown the effect on biomass production (Figure 3C,D, Figure 4 and Figure 5). In general, imidacloprid seems to have a positive effect on yield. Imidacloprid in a 90% and 30% average increase in the total grain yield of the wheat crop raised from seeds treated with the mixture relative to the corresponding untreated control. Imidacloprid seed treatment has important advantages over a conventional spray treatment as it reduces hazards and has a longer period of protection on the irrigated crop [62].

Potato tubers, unlike aerial parts of plants, do not have chloroplasts. Consequently, more energy is required from the mitochondria, and alternative respiratory pathways would interfere with the coupled respiration and ATP synthesis. Indeed, the low activity of alternative oxidase pathways in potato tuber mitochondria was previously shown [63,64]. Consequently, the mitochondria of potato tubers should be less protected from pesticide-induced damage. Indeed, we showed that the imidacloprid+/− and imidacloprid−/+ groups of crops had a minor increased number of mtDNA damage compared with control (Figure 2C). However, the imidacloprid-induced amount of damage in the tubers of the potato crop is significantly less than that induced by damage in vitro (Figure 2C, Table 1). It is noteworthy that the smallest amount of damage was observed in the imidacloprid+/+ group. The mitochondria of tubers of this group also produced the least amount of H_2_O_2_ in comparison with other groups (Figure 7). This looks rather strange, since imidacloprid is an ATPase inhibitor [65]. It is known that the inhibition of ATPase, for example with oligomycin, leads to the hyperpolarization of mitochondria [66] which can lead to an increase in the rate of H_2_O_2_ production [67]. Nevertheless, we observed an inverse relationship. The group that received the most imidacloprid treatment had the lowest rate of H_2_O_2_ production (Figure 7) and the smallest amount of mtDNA damage (Figure 2C). The activity of aconitate hydratase, which may serve as a biosensor for oxidants [68], was also minimal in the imidacloprid+/+ group; however, the differences with other groups were statistically insignificant (Figure 6B). This generally supports data that imidacloprid seed treatment reduces hazards and has a long period of protection on the irrigated crop [62,69,70], which manifested in the reduced level of oxidative stress (Figure 7) and the amount of mtDNA damage (Figure 2C).

So, the effect of pesticides on the integrity of mtDNA in green parts of plants and in crop tubers is insignificant (Figure 2B,C), despite the extremely high toxicity of the studied pesticides in vitro (Figure 2A). Given the effect of pesticides on the integrity of insect [9,10] and mammalian mtDNA [45,52], we can assume that plant mtDNA is more resistant to pesticides than animal mtDNA. This is consistent with the data of This is consistent with the data that herbicide-induced membrane damage (lipid peroxidation) in potato mitochondria was significantly lower than in rat mitochondria. Authors assumed different susceptibility of both kinds of mitochondria with different activities of protective antioxidant agents [71].In general, plants can detoxify the pesticides through multi-specific pathways. The reactions are primarily divided into three canonical phases, including phase I metabolism (pesticide metabolism by cytochrome P_450s_, hydrolases, laccases), phase II conjugation (the metabolic pathway of S—thiols conjugates, detoxification by glutathione S—transferases and glycosyltransferases), and phase III transport (metabolism of pesticides by ATP binding cassette transporters) [72]. We recently showed that plant mitochondria, which oxidized NADH by complex I (not alternative NAD(P)H dehydrogenases), was more prone to pesticide-induced inhibition of the rate of O_2_ consumption [36] than intact mitochondria, which oxidized of succinate [31]. Thus, respiration of plant mitochondria in “animal-like manner” (electron flow from complex I to complex IV) is a factor that makes mitochondria more sensitive to pesticides [36]. In sum, with current research, we can assume that alternative respiratory pathways can also mediate the plant mitochondria resistance to pesticide exposure.

## 4. Materials and Methods

### 4.1. Chemicals

The studied herbicide was 4-amino-6-tert-butyl-3-methylthio-1,2,4-triazine-5(4H)-one (metribuzin, trade name Lazurit (Avgust, Moscow, Russia). The studied insecticides were 4,5-dihydro-N-nitro-1 [(6-chloro-3-pyridyl)-methyl]-imidazolidine-2-ylene-amine (imidacloprid, trade name Tanrek (Avgust, Moscow, Russia); mannitol (Dia-M, Moscow, Russia); sucrose (Dia-M, Moscow, Russia); bovine serum albumin (BSA, Dia-M, Moscow, Russia); dithiothreitol (DTT, Dia-M, Moscow, Russia); ethylenediaminetetraacetic acid (EDTA, Helicon, Moscow, Russia); 3% solution of H_2_O_2_ (Tula pharmaceutical factory, Tula Oblast, Russia); sodium citrate (Sigma Aldrich, St. Louis, MI, USA); Tris-HCl (Dia-M, Moscow, Russia); KOH (Dia-M, Moscow, Russia); HEPES (Bioclot GmbH, Aidenbach, Germany); succinate (Sigma Aldrich, USA); 2,6-dichlorophenolindophenol (DCPIP, Sigma Aldrich, St. Louis, MI, USA); phenazine metasulfate (PMS, Sigma Aldrich, St. Louis, MI, USA); KH_2_PO_4_ (Dia-M, Moscow, Russia); horseradish peroxidase (Sigma Aldrich, St. Louis, MI, USA), AmplexUltraRed (Thermo Fisher Scientific, Waltham, MA, USA); and Oligomycin A (Sigma Aldrich, St. Louis, MI, USA).

### 4.2. Optimization of the Method for Measuring of the Number of mtDNA Damage

The primers panel was developed for six genes of potato mitochondrial DNA (*Solanum tuberosum* L., 1753). The mitochondrial genome of potato was formed by three chromosomes. We examined lesions for two genes from each chromosome: the sites of *26s* rRNA and *cox2* genes for the first chromosome (MN114537.1), *cob* and *cox1* for the second (MN114538.1), and *ccmB* and *sdh3* for the third chromosome (MN114539.1). The primer sequences are shown in Table 4.

The amount of mtDNA damage was assessed by long-range PCR using Encyclo polymerase. The method used was based on the assumption that DNA lesions, such as single-strand breaks, slow down accumulation of the PCR product due to the inhibition of elongation process. Therefore, the rate of PCR product accumulation would be inversely proportional to the number of damaged DNA molecules [44]. The primers were selected based on our earlier results, showing that the optimal length of the PCR product is about 2 kb [45]. To assess the amount of lesions, we compared the ΔCq values of the control and experimental long fragments with the ΔCq values of the control and experimental short fragments, which were used to normalize the amount of mtDNA.

To optimize the panel of primers, mitochondria from potato tubers were isolated according to the method described below. Intact mitochondria were divided into two tubes. The first part of intact mitochondria (control) was incubated with a respiration substrate (10 mM succinate) only. The second part was additionally treated with 500 μM of H_2_O_2_ for 30 min according to our previous research [45]. Then, mtDNA was isolated from both tubes with mitochondria for subsequent measurement of the amount of mtDNA by qPCR. The specificity of the PCR product was assessed by agarose gel electrophoresis in the 2% agarose gel with TAE buffer.

### 4.3. Designs of Experiments

Experiment 1. In vitro experiment (Figure 8A). Intact mitochondria were obtained from potato tubers. The resulting final pellet was purified from supernatant and divided into three tubes. All tubes contained the respiration substrate (10 mM succinate). The isolation medium was added to the first tube (control). A medium containing 200 μM of metribuzin was added to the second tube. A medium containing 200 μM of imidacloprid was added to the third tube. It was previously shown that 200 μM of these pesticides caused the maximal in vitro effect for mitochondrial parameters [28]. The tubes were incubated at room temperature for 30 min, after which mtDNA was isolated from mitochondria for subsequent measurement of the amount of mtDNA damage by long-range PCR.

Experiment 2. The effect of pesticides on the vegetative parts of potatoes.

The potatoes were grown in 250 mL peat pots under controlled conditions at t = 24 °C, cycles of 16 h of light/8 h of darkness, and a relative humidity of at least 40%. The potato seeds were divided into four groups (Figure 8B). The first group (control) (*n* = 14) was not exposed to pesticides either before planting or during the growing season.

The seeds of the second group (metribuzin) (*n* = 14) were not treated with pesticides before sowing; however, on the 38th day of the experiment, the vegetative parts of the plants were treated with the herbicide Lazurit. The recommended work solution of Lazurit was 40 L per 100 kg of tubers. The concentration of metribuzin was 700 g/kg. We treated 2 kg of tubers. Thus, 800 mL of Lazurit solution was needed, or 1.82 mg of metribuzin per 2 kg of tubers. We used Lazurit which was three times higher than the manufacturer’s recommended concentration, or 5.46 mg of metribuzin per 2 kg, or 2.73 mg/kg of metribuzin. 

The seeds of the third group (imidacloprid) (*n* = 14) were pretreated with the Tanrek insecticide. The recommended work solution of Tanrek was 1 L per 100 kg of tubers. The concentration of imidacloprid was 200 g/L. We treated of 2 kg of tubers. Thus, 20 mL of Tanrek solution was needed, or 4 mg of imidacloprid per 2 kg of tubers. We used Tanrek which was three times higher than the manufacturer’s recommended concentration, or 12 mg of imidacloprid per 2 kg, or 6 mg/kg of imidacloprid. Vegetative parts of plants were not treated during the growing season.

The seeds of the fourth group (metribuzin–imidacloprid) (*n* = 14) were pretreated with a solution of the insecticide Tanrek at the rate of 6 mg of imidaclopride per 1 kg of seeds. Then, on the 38th day of the experiment, the vegetative parts of the plants were treated with the herbicide Lazurit at a concentration of 2.73 g/kg per 1 kg of seeds. The duration of the experiment from the day of landing was 52 days. The length of the seedlings was analyzed daily.

The technical germination of potato seedlings was calculated using the Formula (1)
G = *n*/N ∗ 100%(1)
where G is germination, *n* is the number of germinated tubers, and N is the total number of tubers.

The germination energy was calculated using the following Formula (2):E = *n*/N ∗ 100%,(2)
where E is the germination energy, *n* is the number of tubers germinated after ½ the term, and N is the total number of tubers.

On the 52nd day, seedlings were excised and frozen at t = −80 °C for subsequent DNA isolation. The mass of the root system was estimated.

Experiment 3. Field experiment.

The field experiment was carried out in 2021 at the town Biryuch, Belgorod region, Russia (50°39′53.2″ N; 38°23′50.4″ E). The mean monthly temperature was +20 °C and +11 °C (daytime and nighttime temperatures, respectively) in May, +25 °C and +16 °C in June, +29 °C and +20 °C in July, and +29 °C and +18 °C in August. The mean precipitation in May was 92 mm, 108 mm in June, 80 mm in July, and 39 mm in August. Potato tubers, which were used as seed tubers, were planted to a depth of 15 cm on 3 May. Landing was carried out in five groups, each of which has its own processing scheme (Figure 8C,D).

Control: seeds were not treated with pesticides.Imidacloprid+/−: seeds were treated with insecticide before planting.Imidacloprid−/+: seeds were not treated with an insecticide before planting, and treated during the vegetation.Imidacloprid+/+: seeds were treated with an insecticide both before planting and during the vegetation.Metribuzin: seeds were treated with a metribuzin before planting.

The imidacloprid+/− and imidacloprid+/+ groups were treated with Tanrek before planting. The recommended work solution of Tanrek was 1 L per 100 kg of tubers. The concentration of imidacloprid was 200 g/L. We treated of 2 kg of tubers. Thus, 20 mL of Tanrek solution was needed, or 4 mg of imidacloprid per 2 kg of tubers. We used Tanrek which was three times higher than the manufacturer’s recommended concentration, or 12 mg of imidacloprid per 2 kg, or 6 mg/kg of imidacloprid.

The metribuzin group was treated with Lazurit before planting. The recommended dose of Lazurit for preparing work solution was 3.33 g/L. The concentration of metribuzin was 700 g/kg of Lazurit. The final metribuzin concentration in work solution was 2.33 g/L. The recommended work solution of Lazurit was 3 L per 100 m^2^ of soil. The area occupied by the group was 9 m^2^. We used Lazurit which was three times higher than the manufacturer’s recommended concentration. Accordingly, it took 2.7 g of Lazurit containing 1.89 g of metribuzin, which was diluted in 500 mL of water. Moreover, the test area was sprayed.

The imidacloprid−/+ and imidacloprid+/+ groups were treated with Tanrek during the vegetation on 18 July. The recommended work solution of Tanrek was 1 mL per 100 m^2^ of soil or 0.2 g of imidacloprid per 100 m^2^. The area occupied by the group was 9 m^2^. We used Tanrek which was three times higher than the manufacturer’s recommended concentration, or 3 mL/100 m^2^. Additionally, 270 µL of Tanrek (54 mg of imidacloprid) was diluted in 500 mL of water. Furthermore, the test area was sprayed. 

The potato crop was harvested on 29 August. The mass of the harvested potato tubers was measured. Moreover, these tubers were used to isolate the mitochondria and to subsequently measure the H_2_O_2_ production rate, enzyme activity, and mtDNA isolation.

### 4.4. Mitochondria Isolation

Mitochondria were isolated from potato tubers in a medium containing 225 mM of sucrose, 100 mM of mannitol, 20 mM of HEPES, 1 mM of EDTA, 1 mM of DTT, and1.5 mg/L of fatty acids free BSA, pH 7.4 (KOH). The tubers of potatoes were homogenized with Dounce homogenizer (Thermo Fisher Scientific, USA) in the medium in a 1:1 ratio at +4 °C. The homogenate was centrifuged for 5 min at 1000× *g*. The resulting supernatant was transferred into a clean centrifuge tube and centrifuged for 5 min at 4000× *g*. The resulting supernatant was separated from the pellet, transferred to the new centrifuge tube, and centrifuged for 15 min at 15,000× *g*. Then, mitochondria were purified by centrifugation in a 17% Percoll gradient for 30 min at 30,000× *g* with a centrifuge acceleration of 6. After centrifugation, the mitochondrial fraction located 1 cm from the bottom (3–4 mL) was collected, diluted with the medium, and centrifuged for 15 min at 18,000× *g*. The pellet was transferred to a 1.5 mL Eppendorf tube and centrifuged for 10 min at 16,000× *g*. The supernatant was removed and the pellet was resuspended in 150 μL of isolation medium.

### 4.5. DNA Isolation

DNA was isolated using the PROBA-GS kit (DNA-Technologies, Moscow, Russia). DNA was isolated from the green part of seedlings and intact potato mitochondria. Next, 1–2 g of potato seedlings were homogenized in 500 μL of “Lysis buffer” from the kit. For DNA isolation from intact mitochondria, the final pellet, which contained mitochondria, was dissolved in the 500 μL of “Lysis buffer” from the kit. Then, 20 μL of the “Sorbent” was added to the tube with the homogenate (for isolation DNA from seedling) or resuspended pellet (for isolation DNA from intact mitochondria). The tube was mixed on a vortex. The resulting mixture was incubated for 20 min at 50 °C, and centrifuged at 16,000× *g* for one minute. The supernatant was removed. Then, 200 µL of “Wash Solution #1” was added to the sorbent, the mixture was vortexed, and then centrifuged at 16,000× *g* for one minute. Similar steps were carried out with “Wash Solution #2” and “ Wash Solution #3”. After removing the supernatant, the pellet in an open tube was dried by incubation at 50 °C for 5 min. Next, 100 µL of the “Eluent solution” was added to the pellet, mixed on a vortex, and incubated for 5 min at 50 °C. The mixture was centrifuged at 16,000× *g* for one minute, and the supernatant from the eluted DNA was collected. The quality and purity of isolated DNA were estimated by electrophoresis using a 2% agarose gel in a TAE buffer.

### 4.6. Measuring of the mtDNA Damage Amount

To measure the amount of mtDNA lesions, the real-time qPCR method was used with a CFX96^TM^ thermocycler (Bio-Rad, Hercules, CA, USA). PCR reaction protocol: preliminary denaturation—3 min, 95 °C; denaturation—95 °C, 10 s; annealing—59 °C, 30 s; elongation—72 °C, 4 min 30 s; 40 cycles. The PCR reaction mixture per sample included 1X qPCRmix-HS SYBR mix, 0.5 μM of forward and reverse primers, 5 ng of DNA, and 14 μL of deionized water. To assess the amount of lesions, we compared the ΔCq values of the control and experimental long fragments with the ΔCq values of the control and experimental short fragments, which were used to normalize the amount of mtDNA. The amount of mtDNA damage was calculated at 10 kb according to Formula (3):
(3)$$D_{10}=\frac{(1-E^{-(\Delta \mathrm{sh}-\Delta L)})*10\;\mathrm{kbp}}{L_{f}}$$
where D_10_—the value of damage per 10,000 base pairs; E—the value of the PCR efficiency; ΔSh = Cq of the short target fragment—Cq of the short control fragment; ΔL = Cq of the long target fragment—Cq of the long control fragment; and L_f_ —the length of the fragment, bp.

### 4.7. Assays of Enzyme Activity

The medium for measuring the activity of aconitate hydratase contained the following components: 20 mM of sodium citrate; 50 mM of Tris-HCl (pH 7.8). The reaction was initiated by the addition of mitochondria. The increase in the optical density was detected at 240 nm due to the formation of the double bond in cis-aconitate [73]. The medium for measuring the activity of succinate dehydrogenase contained the following components: 10 mM of HEPES, 50 mM of Tris-HCl (pH 7.4); 40 mM of succinate; 50 μM of 2,6-DCPIP; 50 μM of PMS; and 20 μM of EDTA. The reaction was initiated by the addition of succinate. The succinate dehydrogenase activity was determined by a decrease in the absorption at 600 nm, caused by reduction in the artificial electron acceptor DCPIP [74]. The measurements were carried out on a Hitachi U-2900 spectrophotometer at a wavelength of 240 nm for aconitate hydratase and 600 nm for succinate dehydrogenase. The measurements were carried out in 1 mL of medium with the addition of 15 μM of mitochondria for 3 min. The enzyme activity was calculated on the basis that the coefficient of molar extinction for aconitate hydratase is 3.08 cm^−1^ * mM^−1^ and that the succinate dehydrogenase is 21 cm^−1^ * mM^−1^. The enzyme activity is presented in mmol/min/g of fresh weight (FW). 

### 4.8. Assays of H_2_O_2_ Rate Production

The assay of H_2_O_2_ production rate was recorded using Amplex Red Ultra fluorescent dye. The measurements were carried out on a Hitachi F-7000 spectrofluorometer at an extinction wavelength of 568 nm and an emission wavelength of 581 nm. The assay of H_2_O_2_ production rate was carried out in an acrylic cuvette in the medium contained 225 mM of sucrose, 100 mM of mannitol, 20 mM of HEPES; 1 mM E of DTA; 1.5 mg/L of BSA free from fatty acids, pH 7.4; 1 mL of medium; 10 mM of succinate as a respiration substrate; 4 mM of KH_2_PO_4_; four units of horseradish peroxidase; and 1 μM of Amplex Ultra Red. Then, 15 μL of mitochondria was added to the cuvette. Moreover, for an assay of maximal rate of H_2_O_2_ production, 10 μM of oligomycin was added. Values of the rate of H_2_O_2_ production were recorded for mitochondria, before oligomycin adding and after oligomycin adding. The concentration of H_2_O_2_ was measured as the fluorescence intensity of the resorufin formed during the reaction [75]. The protein concentration was measured using the PierceTM BCA Protein Assay Kit (Thermo Fisher Scientific, USA).

### 4.9. Statistical Analysis

Statistical analysis was performed using the Statistica 12 software package (StatSoft. Inc, Tulsa, OK, USA). DNADamageCalculator (Voronezh, Russia) was used for the quantification of the number of mtDNA damage. The distribution normality was determined using the Shapiro–Wilk test. The data were represented as the median (Q1, Q3). The results were analyzed by one-way analysis of variance (ANOVA). Tukey’s post-hoc test was used to determine the significance level.

## 5. Conclusions

Thus, our studies have shown that the use of pesticides, such as metribuzin and imidacloprid, does not damage potato mitochondria. At the same time, it should be taken into account that pesticides can accumulate in water and soil, and can harm animals and humans. Agricultural workers who mix, load, transport, and directly use pesticides are generally considered to be the group that will receive the most exposure and, therefore, are most at risk [76]. Expanding understanding of the mechanism of plant protection from pesticide-induced toxicity can help in the development of pharmacological methods for protecting humans and animals from the negative effects of pesticides. Together, this and our previous research [35] suggest that alternative respiratory pathways of plant mitochondria, which provide uncoupled respiration, may be responsible for plant resistance to pesticides. The pharmacological pathway for the uncoupling of mitochondrial respiration in animal mitochondria may be a perspective approach to help develop methods for protecting agricultural workers from the harmful effects of pesticides.

## Figures and Tables

**Figure 1 ijms-23-02970-f001:**
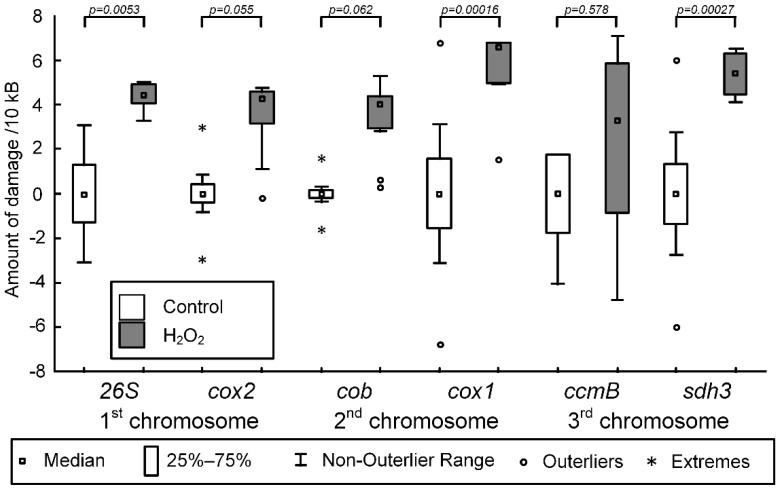
The amount of damage per 10 kb of mtDNA induced by 500 μM H_2_O_2_ for each of the six fragments. *26s* and *cox2* are fragments of the 1st chromosome. *Cob* and *cox1* are fragments of the 2nd chromosome. *CcmB* and *sdh3* are fragments of the 3rd chromosome. Significance levels between control and H_2_O_2_-treated groups are presented. The *p* value corresponds to Tukey’s post-hoc test. *p* < 0.05 was considered to be statistically significant.

**Figure 2 ijms-23-02970-f002:**
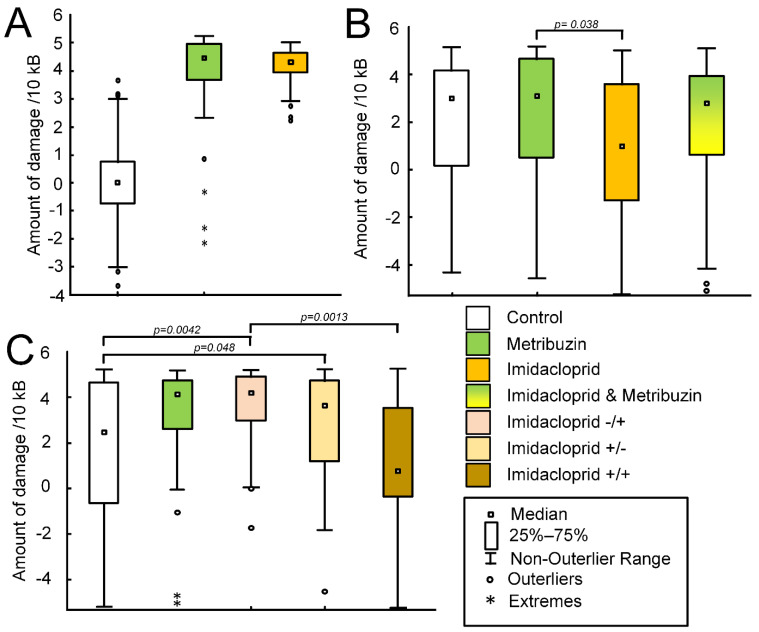
The amount of damage per 10 kb of mtDNA induced by pesticides. (**A**) The amount of mtDNA damage, induced by incubation of intact mitochondria for 30 min with respiratory substrate and pesticides (metribuzin or imidacloprid). The control mitochondria were incubated with respiratory substrate only. (**B**) The amount of mtDNA damage in potato shoots after treatment by metribuzin and imidacloprid, and co-treatment by metribuzin and imidacloprid. (**C**) The amount of mtDNA damage in potato tubers obtained at harvest in a field experiment. The control group did not treat any pesticides. The metribuzin group: seeds were treated herbicide Lazurit before planting only. Imidacloprid−/+ group: seeds were not treated before planting, and treated with Tanrek during the vegetation. The imidacloprid+/− group: seeds were treated with Tanrek before planting only. The imidacloprid+/+ group: seeds were treated with Tanrek both before planting and during the vegetation. The *p* value corresponds to Tukey’s post-hoc test. *p* < 0.05 was considered to be statistically significant.

**Figure 3 ijms-23-02970-f003:**
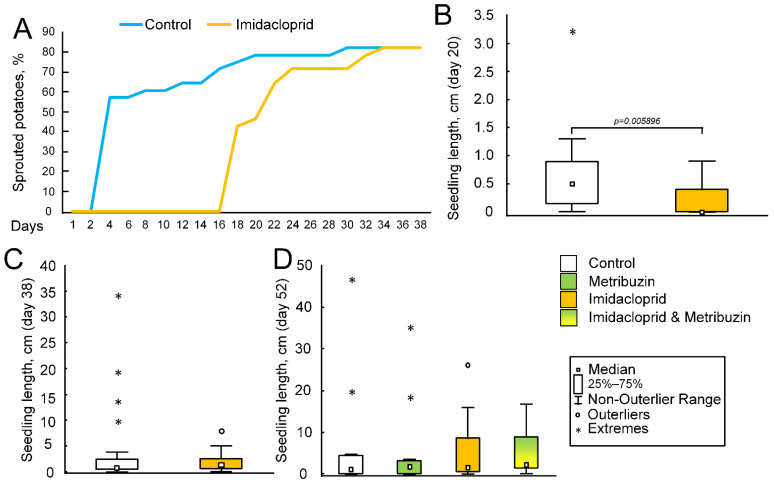
Seed germination and potato growth under controlled conditions. (**A**) Dynamic germination of potato seeds. (**B**) Influence of imidacloprid treatment on seedling length at day 20. The *p* value corresponds to Tukey’s post-hoc test. *p* < 0.05 was considered to be statistically significant. (**C**) The influence of imidacloprid treatment on seedling length on day 38. There was no statistically significant difference. (**D**) The influence of metribuzin and imidacloprid treatment on seedling length at day 52. There were no statistically significant differences.

**Figure 4 ijms-23-02970-f004:**
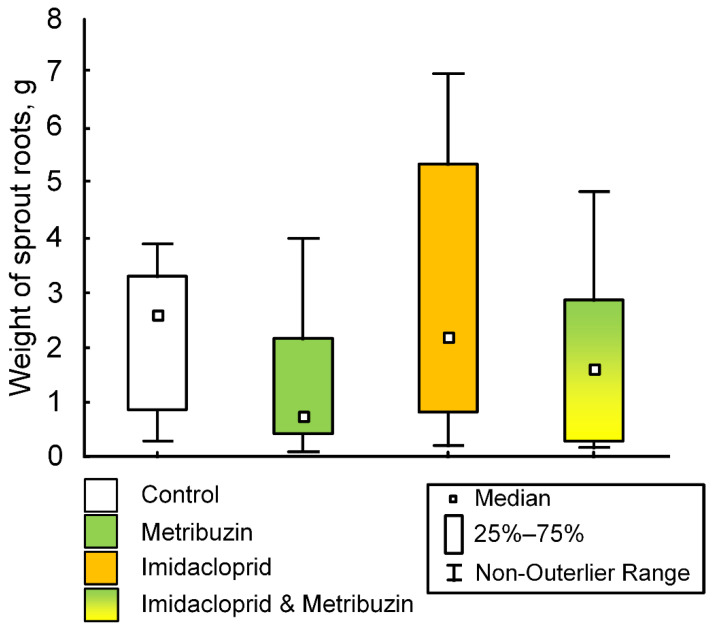
The weight of root system of sprouts at day 52 after treatment by metribuzin and imidacloprid, and co-treatment by metribuzin and imidacloprid. There are no statistically significant differences.

**Figure 5 ijms-23-02970-f005:**
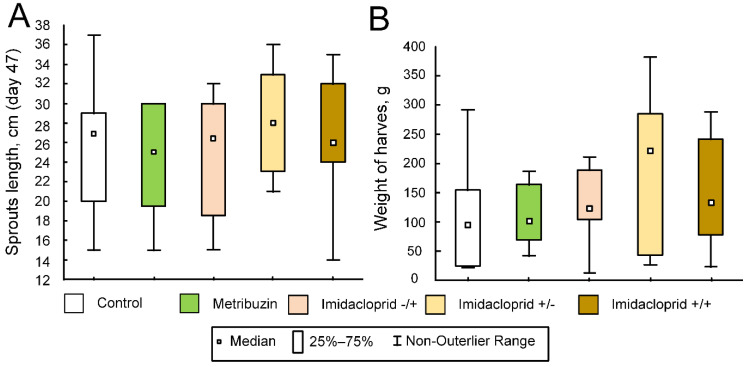
The effect of pesticides on potato growth and crop weight in the field experiment. The control group was not treated with any pesticides. The metribuzin group: seeds were treated with herbicide Lazurit before planting only. The imidacloprid−/+ group: seeds were not treated before planting, and were treated with Tanrek during the vegetation. The imidacloprid+/− group: seeds were treated with Tanrek before planting only. The imidacloprid+/+ group: seeds were treated with Tanrek both before planting and during the vegetation. (**A**) The length of sprouts at day 52. (**B**) The weight of potato harvest at the end of experiment. There were no statistically significant differences.

**Figure 6 ijms-23-02970-f006:**
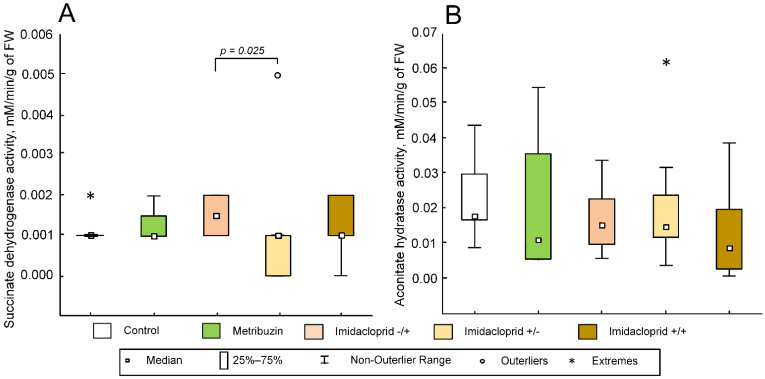
Activity of succinate dehydrogenase (**A**) and aconitate hydratase (**B**) in mitochondria from potato tubers. The control group was not treated with any pesticides. The metribuzin group: seeds were treated with herbicide Lazurit before planting only. The imidacloprid−/+ group: seeds were not treated before planting, and were treated with Tanrek during the vegetation. The imidacloprid+/− group: seeds were treated with Tanrek before planting only. The imidacloprid+/+ group: seeds were treated with Tanrek both before planting and during the vegetation. The *p* value corresponds to Tukey’s post-hoc test. *p* < 0.05 was considered to be statistically significant.

**Figure 7 ijms-23-02970-f007:**
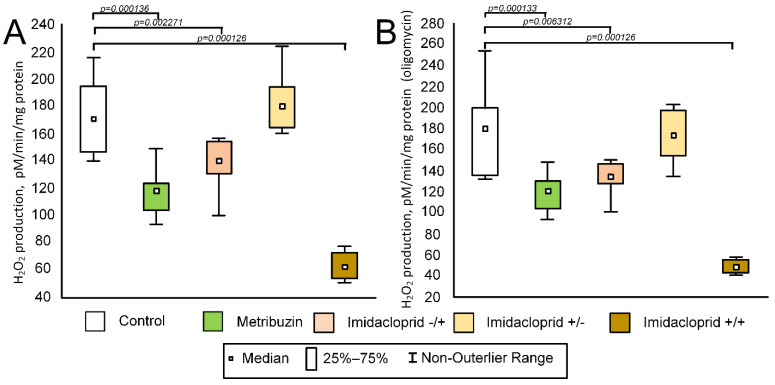
(**A**) Rate of H_2_O_2_ production in mitochondria of potato tubers without oligomycin. (**B**) Rate of H_2_O_2_ production in oligomycin-inhibited mitochondria of potato tubers. The control group was not treated with any pesticides. The metribuzin group: seeds were treated with herbicide Lazurit before planting only. The imidacloprid−/+ group: seeds were not treated before planting, and were treated with Tanrek during the vegetation. The imidacloprid+/− group: seeds were treated with Tanrek before planting only. The imidacloprid+/+ group: seeds were treated with Tanrek both before planting and during the vegetation. The *p* value corresponds to Tukey’s post-hoc test. *p* < 0.05 was considered to be statistically significant.

**Figure 8 ijms-23-02970-f008:**
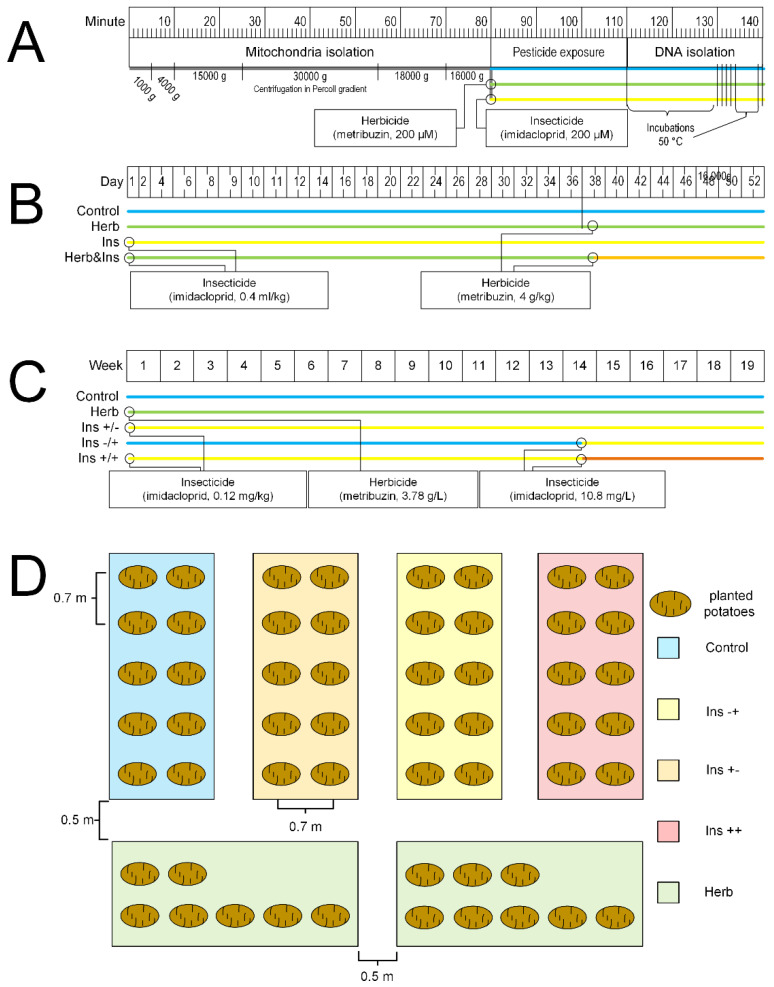
Schemes of in vitro experiment (**A**), study of the effect of pesticides on the vegetative parts of potato under controlled conditions (**B**), field experiment (**C**), and schedule of potato planting (**D**). (**A**) In vitro experiment. Mitochondria isolation for purifying mitochondria samples. Furthermore, pesticide exposure continued for 30 min. The concentration of each pesticide was 200 μM. In addition, DNA isolation was used to estimate the number of pesticide-induced damage. (**B**) Study of the effect of pesticides on the vegetative parts of potato under controlled conditions. Potato shoots were treated by metribuzin and imidacloprid, and co-treated by metribuzin and imidacloprid. (**C**,**D**) Field experiment. The control group did not treat any pesticides. The metribuzin group: seeds were treated herbicide Lazurit before planting only. The imidacloprid−/+ group: seeds were not treated before planting, and treated with Tanrek during the vegetation. The imidacloprid+/− group: seeds were treated with Tanrek before planting only. The imidacloprid+/+ group: seeds were treated with Tanrek both before planting and during the vegetation. *p* values correspond to Tukey’s post-hoc test. *p* < 0.05 was considered to be statistically significant.

**Table 1 ijms-23-02970-t001:** The number of mtDNA damage in intact potato mitochondria in each fragment.

mtDNA Fragment	Control	Metribuzin	Imidacloprid
*26s*	0.00 (−1.57; 1.57)	4.35 (3.56; 4.48) ^11^	4.26 (3.95; 4.59) ^22^
	F (2,17) = 20.6552, *p* = 0.00003		
*Cox2*	0.00 (−1.59; 1.59)	4.58 (3.58; 4.77) ^1^	4.56 (4.14; 4.70) ^22^
	F (2,17) = 10.1543, *p* = 0.0013		
*Cob*	0.00 (−1.83; 1.83)	4.01 (2.65; 4.28)	4.39 (4.15; 4.58) ^22^
	F (2,17) = 6.1623, *p* = 0.0097		
*Cox1*	0.00 (−0.75; 0.75)	4.8 (4.28; 5.03) ^111^	4.55 (4.03; 4.78) ^22^
	F (2,17) = 39.88, *p* = 0.00001		
*ccmB*	0.00 (−1.5;1.5)	4.72 (3.89; 5.23) ^111^	4.22 (3.63; 4.38) ^22^
	F (2,17) = 17.0728, *p* = 0.00009		
*Sdh3*	0.00 (−1.11:1.11)	4.96 (3.84; 5.10) ^1^	4.24 (3.80; 4.51) ^22^
	F (2,17) = 6.5538, *p* = 0.0078		

The *p* value corresponds to Tukey’s post-hoc test. ^1^ *p* < 0.05 offers a comparison between the metribuzin and control groups. ^11^ *p* < 0.01 offers a comparison between the metribuzin and control groups. ^111^ *p* < 0.001 offers a comparison between the metribuzin and control groups. ^22^ *p* < 0.01 offers a comparison between the imidacloprid and control groups.

**Table 2 ijms-23-02970-t002:** The amount of mtDNA damage in potato shoots.

mtDNA Fragment	Control	Metribuzin	Imidacloprid	Metribuzin–Imidacloprid
*26s*	4.34 (2.04; 4.81)	1.64 (0.11; 2.71) ^111^	1.21 (−1.31; 3.22) ^22^	1.96 (−0.88; 2.99) ^333^
	F (3,116) = 3.99, *p* = 0.0095			
*cox2*	3.42 (1.71; 4.21)	2.86 (0.92; 3.62)	3.28 (2.28; 3.87)	2.69 (0.40; 3.62)
	F (3,120) = 1.108, *p* = 0.35			
*cob*	2.025 (−0.67; 3.66)	4.43 (3.29; 4.73)	1.53 (−0.99; 3.59) ^44^	2.75 (−0.42; 4.18)
	F (3,116) = 3.99, *p* = 0.0095			
*cox1*	−1.49 (−4.10; 3.56)	1.40 (−3.80; 4.8)	−2.28 (−4.35; 4.85)	3.76 (−2.81; 4.76)
	F (3,108) = 1.232, *p* = 0.3			
*ccmB*	3.45 (−1.22; 4.49)	4.2 (1.43; 4.79)	−5.11 (−5.23; 3.95) ^22444^^555^	4.43 (1.09; 4.83) ^666^
	F (3,104) = 7.2946, *p* = 0.0002			
*sdh3*	4.80 (2.48; 4.97)	4.76 (4.60; 5.10) ^1^	4.49 (2.59; 4.80)	4.65 (0.84; 4.89) ^5^
	F (3,108) = 3.2145, *p* = 0.026			

The *p* value corresponds to Tukey’s post-hoc test. ^1^ *p* < 0.05 offers a comparison between the metribuzin and control groups. ^111^ *p* < 0.001 offers a comparison between the metribuzin and control groups. ^22^
*p* < 0.01 offers a comparison between the imidacloprid and control groups. ^333^
*p* < 0.001 offers a comparison between the metribuzin–imidacloprid and control groups. ^44^
*p* < 0.01 offers a comparison between the imidacloprid and metribuzin groups. ^444^
*p* < 0.001 offers a comparison between the imidacloprid and metribuzin groups. ^5^ *p* < 0.05 offers a comparison between the imidacloprid–metribuzin and metribuzin groups. ^555^
*p* < 0.001 offers a comparison between the imidacloprid and metribuzin groups. ^666^
*p* < 0.001 offers a comparison between the imidacloprid–metribuzin and imidacloprid groups.

**Table 3 ijms-23-02970-t003:** The amount of mtDNA damage in potato tubers.

mtDNA Fragment	Control	Metribuzine	Imidacloprid−/+	Imidacloprid+/−	Imidacloprid+/+
*26s*	3.26 (−4.89; 5.00)	4.43 (2.88; 5.00)	4.16 (3.68; 4.45)	1.54 (−0.92; 3.00) ^888^	1.31 (−1.17; 3.18) ^99^
	F (4,123) = 6.22, *p* = 0.0001				
*cox2*	3.59 (−4.67; 4.73)	4.70 (4.00; 4.79)	4.29 (3.52; 4.77)	3.10 (0.00; 4.69)	3.74 (−0.12; 4.72)
	F (4,123) = 2.37, *p* = 0.056				
*cob*	4.72 (4.68; 4.73)	4.70 (−4.70; 4.73)	2.56 (1.25; 4.66)	4.13 (3.17; 4.73)	4.73 (4.68; 4.74) ^7^
	F (4,91) = 4.58, *p* = 0.0021				
*cox1*	1.69 (−4.78; 4.31)	2.77 (−5.03; 4.92)	4.93 (3.73; 5.01) ^22^	4.90 (4.50; 5.00) ^33^	−0.33 (−3.25; 2.79)
	F (4,115) = 8.77, *p* = 0.00001				
*ccmB*	4.61 (−3.62; 5.17)	4.60 (3.15; 5.07)	4.82 (3.44; 5.10)	4.71 (1.72; 5.13)	0.00 (−4.78; 4.45) ^777999^***
	F (4,123) = 7.10, *p* = 0.00004				
*sdh3*	4.35 (−4.45; 4.93)	4.05 (3.40; 4.40)	3.77 (2.25; 4.56)	4.71 (1.72; 5.13)	−3.84 (−5.07; 1.99) ^4777999^***
	F (4,23) = 11.89, *p* = 0.00001				

The *p* value corresponds to Tukey’s post-hoc test. ^22^ p < 0.01 offers a comparison between the imidacloprid−/+ and control groups. ^33^
*p* < 0.01 offers a comparison between the imidacloprid +/− and control groups. ^4^
*p* < 0.05 offers a comparison between the imidacloprid+/+ and control groups. ^7^ *p* < 0.05 offers a comparison between the imidacloprid+/+ and metribuzin groups. ^777^ *p* < 0.001 offers a comparison between the imidacloprid+/+ and metribuzin groups.^888^
*p* < 0.001 offers a comparison between the imidacloprid−/+ and imidacloprid+/− groups. ^99^
*p* < 0.01 offers a comparison between the imidacloprid−/+ and imidacloprid+/+ groups. ^999^
*p* < 0.001 offers a comparison between the imidacloprid−/+ and imidacloprid+/+ groups. *** *p* < 0.001 offers a comparison between the imidacloprid+/− and imidacloprid+/+ groups.

**Table 4 ijms-23-02970-t004:** The characteristics of used primers.

Chromosome	Gene	Sequence(5′-3′)	PCR-Product	Fragment Length (bp)	Percent of Gene Coverage
1	*26s* Forward	CATCCGCCCCAGATAAACTA			
	*26s* Reverse Long	GTACCGTGAGGGAAAGGTGA	33993-35978	1986	57
	*26s* Reverse Short	ATAGGTGGGAGGTGGTGACA	33993-34126	134	4
	*cox2* Forward	CGTACAGGGAAGGTGGAAGA			
	*cox2* Reverse Long	TTTAAACGACCCGGTACAGC	212390-214469	2080	96
	*cox2* Reverse Short	CACAAGGAGCAATTGTGAGG	212390-212510	121	5
2	*cob* Forward	TTGACATCCGATCCAACCTA			
	*cob* Reverse Long	AATGTAACTCCCGAAGGA	33662-35769	2108	88
	*cob* Reverse Short	CCATGCCATTCTTCGTAGTA	33662-33854	193	16
	*cox1* Forward	ATAATCTGGAATGCGACGTG			
	*cox1* Reverse Long	TAAACCCAATGGGAACCAAA	76578-78565	1988	83
	*cox1* Reverse Short	GGACATACCCTGAAACTTTA	76578-76699	122	8
3	*ccmB* Forward	AACGCCCTTAATGCTAGGTT			
	*ccmB* Reverse Long	CCCCCTCCCGCTATTACTAC	10037-11945	1909	85
	*ccmB* Reverse Short	TTTGGCAAGCAATAAGCACT	10037-10201	165	27
	*sdh3* Forward	CCCTATCTCCTCATCTTCCT			
	*sdh3* Reverse Long	GTGGCTCGTCCGTGATAACT	35468-37421	1936	74
	*sdh3* Reverse Short	AGGTGAAGCAAATCAAACCT	35468-35634	149	46

## Data Availability

Data are available from the corresponding authors upon request.

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
