# Peer review of "High Doses of Pesticides Induce mtDNA Damage in Intact Mitochondria of Potato In Vitro and Do Not Impact on mtDNA Integrity of Mitochondria of Shoots and Tubers under In Vivo Exposure"

_ijms, 2022, doi:10.3390/ijms23062970_

Round 1
Reviewer 1 Report
Dear Editors,
Thank you so much for choosing me as a editor of the manuscript ID ijms-1592949 entitled “The effect of pesticides exposure on the potato mitochondrial functions and mitochondrial DNA integrity” submitted to International Journal of Molecular Sciences. I hope that my comment will help Authors to improve their manuscript.
Detailed remarks concerning manuscript.
- All the Latin names of the plants and animals species should be italicized (see Solanum tuberosum page 1 Line 13; bumblebees (Bombus terrestris) page 19 Line 657). The same concerns phrase in vitro (page 1 line 14). Please do needed changes in the whole text of the manuscript.
- Not only the clear goal, but also the scientific hypothesis of the studies should be given together with the clear answer for the question stated as a scientific hypothesis presented in the conclusions.
- The clear practical use for the results of the studies as well as the recommendation for the future studies should be presented.
- The clear conclusions based on the obtained studies should be given.
- Key words. It is not recommended to use as key words the words or phrases used in the title of the manuscript. Please do needed changes.
- Bocharov and Fomin showed that insecticides Kruiser (thiamethoxam) and Prestige (imidacloprid, pencycuron), in a mixture with the fungicide Maxim, fludioxonil), inhibited growth processes at the initial stage of ontogenesis (germination - seedlings). (Page 13 lines 389-392). The reference Bochoarov and Fomin seems is not to be cited in the reference list. Please double check whether all citations in the text of the manuscript are mentioned in reference list and vice versa. The way of the citation should be strictly such as that presented in the guides for authors.
- “Second part was additionally treated with a 500 μM of H2O2 according to (Gureev et al. 2017) for 30 minutes [43].” (Page 15 lines 471-472). Why the double citation of the references is used?
- I suggest to divide the “Discussion” section to the subsections corresponded to the subsections for section “Results” and discuss all studied aspects.
- Reference list contains many editorial mistakes. It is impossible to mention all of them. There are only few examples. Once the journal titles are abbreviated, but the other time not. Once each word in the manuscript tile is written with capital letter, but the other time only first word of the manuscript is written with capital letter. Please go through the whole reference list and do needed changes. Reference list should be prepared according to the one scheme presented in the guides for authors.
Author Response
Response to Reviewer #1.
Thank you very much for your valuable comments. We took your comments into consideration and made revision. We present below the list of corrections and responses to your questions.
- “All the Latin names of the plants and animals species should be italicized (see Solanum tuberosum page 1 Line 13; bumblebees (Bombus terrestris) page 19 Line 657). The same concerns phrase in vitro (page 1 line 14). Please do needed changes in the whole text of the manuscript.”
We fixed Latin names and concerns phrase in the whole text of the manuscript.
- “Not only the clear goal, but also the scientific hypothesis of the studies should be given together with the clear answer for the question stated as a scientific hypothesis presented in the conclusions”.
Thank you very much for this remark. We added our scientific hypothesis that in vivo plant mitochondria may be significantly more resistant that intact plant mitochondria under pesticides exposure. We also rewrote goal of research.
- “The clear practical use for the results of the studies as well as the recommendation for the future studies should be presented.”
We added recommendation for the future studies in the conclusion section.
- “The clear conclusions based on the obtained studies should be given.”
We added conclusion. The conclusion summarizes our findings and the data of our previous studies. Recommendation for the future studied was also added.
- “Key words. It is not recommended to use as key words the words or phrases used in the title of the manuscript. Please do needed changes.”
Thank you for this recommendation. We update key words.
- “Bocharov and Fomin showed that insecticides Kruiser (thiamethoxam) and Prestige (imidacloprid, pencycuron), in a mixture with the fungicide Maxim, fludioxonil), inhibited growth processes at the initial stage of ontogenesis (germination - seedlings). (Page 13 lines 389-392). The reference Bochoarov and Fomin seems is not to be cited in the reference list. Please double check whether all citations in the text of the manuscript are mentioned in reference list and vice versa. The way of the citation should be strictly such as that presented in the guides for authors.”
We carefully checked that citations in the text of the manuscript are mentioned in reference list. Citations were formatted according to the guides for authors.
- “Second part was additionally treated with a 500 μM of H2O2 according to (Gureev et al. 2017) for 30 minutes [43].” (Page 15 lines 471-472). Why the double citation of the references is used?”
We fixed this mistake.
- “I suggest to divide the “Discussion” section to the subsections corresponded to the subsections for section “Results” and discuss all studied aspects.”
Thank you for this suggestion. We divided Discussion to the subsections. However, the structure of Discussion section does not repeat the structure of the Results section.
- “Reference list contains many editorial mistakes. It is impossible to mention all of them. There are only few examples. Once the journal titles are abbreviated, but the other time not. Once each word in the manuscript tile is written with capital letter, but the other time only first word of the manuscript is written with capital letter. Please go through the whole reference list and do needed changes. Reference list should be prepared according to the one scheme presented in the guides for authors.”
Reference list was formatted according to the rules of the journal
Thank you very much for your valuable remarks, which helped us to improve our manuscript.
Reviewer 2 Report
1. The work is very interesting. Potatoes are one of the main sources of starch in the diet. Changes that occur at the DNA level under the influence of pesticides are important to be studied. The work is well planned and structured. The experimental plan is well chosen. I have no comments on the experimental part.
2. However, referring to the information on industrial production, there was no data on the impact on human health. We all agree that science serves people and should take care of their health and life, not only the financial aspect. The work should contain a description of the impact of the authors' research on human health. Will the changes noted by the authors have a positive effect on health (e.g. digestive / immune systems)?
Author Response
Response to Reviewer #2.
“The work is very interesting. Potatoes are one of the main sources of starch in the diet. Changes that occur at the DNA level under the influence of pesticides are important to be studied. The work is well planned and structured. The experimental plan is well chosen. I have no comments on the experimental part. However, referring to the information on industrial production, there was no data on the impact on human health. We all agree that science serves people and should take care of their health and life, not only the financial aspect. The work should contain a description of the impact of the authors' research on human health. Will the changes noted by the authors have a positive effect on health (e.g. digestive / immune systems)?”
We thank you for your appropriate and important comment. Agricultural workers who mix, load, transport and directly use pesticides are generally considered to be the group that will receive the most exposure and therefore are most at risk. Expanding understanding of the mechanism of plant protection from pesticide-induced toxicity can help in the development of pharmacological methods for protecting humans and animals from the negative effects of pesticides. Together, this and our previous research suggest that alternative airways of plant mitochondria, which provide uncoupled respiration, may be responsible for plant resistance to pesticides. Pharmacological pathway for uncoupling of mitochondrial respiration in animal mitochondria may be perspective approach for development of methods for protection of agricultural workers from the harmful effects of pesticides. We added this information to conclusion. Thank you for your comment.
Reviewer 3 Report
The manuscript by Alimova et al. describes experiments on pesticide effects on mitochondria in potato.
The English language of this paper makes it scarcely understandable at times, and also the lack of scientific language and appropriate structure is evident.
The explanations of results are redundant with figures. Figures are not of sufficient quality and legends are not sufficient to understand their content.
The introduction and the results are not clear about the use of "pesticide" in the sense of herbicides and insecticides. The products used for the experiments are not mentioned until line 339 of the manuscript. The fact that the herbicide has no toxic effect on potato plants should need an explanation.
The method for determining and quantifying the mtDNA damage is not described in sufficient details and therefore all the related results are not understandable.
Most methods are not described with sufficient detail.
The authors are using measure units which are not correct for molarity and volumes
The discussion is too speculative and not supported by the results shown
Understanding this paper is a problem. I have made some comments below but the paper needs a complete rewriting
The title is not informative - it seems the title of a review paper
The abstract is not informative, with vague claims and no information for readers
The keywords overlap with title and abstract and are not informative
L35-44 these claims are not substantiated by data and the writing is not scientific
L46 and others: the authors are mixing together insecticides, fungicides, herbicides without any reference to differences in mechanisms of action
L52 pesticides are not "toxic" for air water and soil
L75 with no indication about "the recommended dose" this has no meaning
L72-78 the purpose of the paper is therefore purely descriptive with no hypothesis?
L81 Considering that in this paper the methods come after the results, it is necessary to provide some information to the reader. The results begin with no explanation of what the authors are doing and why - instead all the explanations are in the discussion
Figure 1 - the quality of this figure is not acceptable for an international publication
L91-100 this paragraph cannot be understood - what are the first, second and third chromosomes? Moreover, the description repeats the numbers visible in the figure and so it is not useful
L109 it should be explained which herbicide and which insecticide
L172 does it mean that the insecticide protects mtDNA from damage? decrease compared to what?
-no more comments on results -
L294-311 this explanation should have been given at the beginning
L323-325 wrong format of references, missing from the final list
L328-329 I did not find these data in the results
L339-346 this should not be in the discussion
Author Response
Response to Reviewer #3.
Thank you very much for your valuable comments. We took your comments into consideration and made a major revision. Below we presented the list of corrections and responses to your questions.
- “The English language of this paper makes it scarcely understandable at times, and also the lack of scientific language and appropriate structure is evident.”
Thank you very much for this comment. We significantly rewrote the manuscript. Spell-checking was provided.
- The explanations of results are redundant with figures. Figures are not of sufficient quality and legends are not sufficient to understand their content.
We changed figures significantly. We removed gel photo due the low resolution of this figure. We combined data of the number of mtDNA damage from three to one figure (use only data average from all chromosome). We transferred the data for each mtDNA fragment to tables. Thus, we reduced the number of figures from 11 to 8. We fixed legends and figure captures in order to improve the perception of data in the figures.
- The introduction and the results are not clear about the use of "pesticide" in the sense of herbicides and insecticides. The products used for the experiments are not mentioned until line 339 of the manuscript. The fact that the herbicide has no toxic effect on potato plants should need an explanation.
We used the term “pesticide” where we meant both herbicide and insecticide. We agree that the name of the herbicide and insecticide was mentioned for the first time in the methods, at the end of the manuscript, which makes it difficult to understand. Therefore, throughout the text we have replaced “herbicides” and “insecticides” to “metribuzin” and “imidacloprid”.
- The method for determining and quantifying the mtDNA damage is not described in sufficient details and therefore all the related results are not understandable.
We added description of principle of method of long-range PCR for estimation the number of mtDNA damage in the chapter 5.2. The amount of mtDNA damage was calculated per 10 kb of mtDNA. We added this measure units to results sections, where number of mtDNA damage was mentioned.
- Most methods are not described with sufficient detail.
We added information about principle of method of long-range PCR enzymes activity assays, H2O2 rate production measurement, expanded description of mitochondria and DNA isolation. We added recommended dose of used pesticides to methods.
- The authors are using measure units which are not correct for molarity and volumes
We fixed these mistakes.
- The discussion is too speculative and not supported by the results shown
Thank you very much for this remark. We replaced part numbers of discussion to introduction section. For example, description of potato mitogenome structure, studied pesticides et al. The discussion in the revised version of manuscript is mostly focused on the obtained results.
- Understanding this paper is a problem. I have made some comments below but the paper needs a complete rewriting
Thank you very much for your valuable comments. We took your comments into account and conducted the major revision of manuscript.
- The title is not informative - it seems the title of a review paper
We changed title of paper. Revised version of title is not descriptive and reflects main results of research
- The abstract is not informative, with vague claims and no information for readers
We rewrote the abstract. We focused on the obtained results and its relation to the scientific hypothesis that we formulated in the goal of the study in the introduction.
- The keywords overlap with title and abstract and are not informative
Thank you for this recommendation. We update key words.
- L35-44 these claims are not substantiated by data and the writing is not scientific
We agree that these sentences are excess and removed it. We indicated only that widespread use of pesticides was the main reasons for increase in yield of potato.
- L46 and others: the authors are mixing together insecticides, fungicides, herbicides without any reference to differences in mechanisms of action
We clarified that most of insecticides and fungicides are mitochondria-targeted. Herbicides, as a rule, inhibits electron flow within the photosynthetic electron transport chain but can cause mitochondrial disfunction too
- L52 pesticides are not "toxic" for air water and soil
Thank you very much for this remark. Indeed, pesticides can pollute water and soil, but are not “toxic” for them. However, this information is excess and we deleted it from introduction.
- L75 with no indication about "the recommended dose" this has no meaning
We removed mentioning of recommended dose in the revised version of the introduction. We added the value of the recommended dose in the material and methods section.
- L72-78 the purpose of the paper is therefore purely descriptive with no hypothesis?
Thank you very much for this remark. We added our scientific hypothesis that in vivo plant mitochondria may be significantly more resistant that intact plant mitochondria under pesticides exposure. We also rewrote goal of research.
- L81 Considering that in this paper the methods come after the results, it is necessary to provide some information to the reader. The results begin with no explanation of what the authors are doing and why - instead all the explanations are in the discussion
We added explanation why we used H2O2 for optimization of method of estimation of number of mtDNA damage.
- Figure 1 - the quality of this figure is not acceptable for an international publication
Unfortunately, we have not gel photo with higher resolution. We removed figure 1 from paper taking into account your previous remark that the explanations of results are redundant with figures. The gel photo did not have data that could affect the conclusions that we present in the paper, it only confirmed the validity of the data, which we presented.
- L91-100 this paragraph cannot be understood - what are the first, second and third chromosomes? Moreover, the description repeats the numbers visible in the figure and so it is not useful
We replaced description of potato mitogenome from discussion to introduction.
- L109 it should be explained which herbicide and which insecticide
We replaced “herbicides” and “insecticides” to “metribuzin” and “imidacloprid” through the whole manuscript.
- L172 does it mean that the insecticide protects mtDNA from damage? decrease compared to what?
We meant that insecticide (imidacloprid) caused a smaller number of mtDNA damage in potato shoots than herbicide (metribuzin). Thus, we do not claim that imidacloprid has a protective effect. We have shown that imidacloprid was less toxic for shoots mtDNA than metribuzin.
- L294-311 this explanation should have been given at the beginning
We moved the explanation of structure of potato mitogenome to the introduction.
- L323-325 wrong format of references, missing from the final list
We carefully checked that citations in the text of the manuscript are mentioned in reference list. Citations were formatted according to the guides for authors.
- L328-329 I did not find these data in the results
We moved these data into the results section. Explanation of the results of the correlation analysis was left in the discussion section
- L339-346 this should not be in the discussion
We moved this section from the introduction.
Thank you very much again for the detailed analysis of our manuscript. We believe that we have been able to significantly improve the perception of the results thanks to your comments.
Round 2
Reviewer 2 Report
Accept in present form
Author Response
We are grateful to you for the high evaluation of our manuscript.
Sincerely, authors.
Reviewer 3 Report
The paper has been improved. Some language problems still remain, for instance in Figure 5,6,7 legend. More work is needed on this.
The movement to introduction of the part on mitogenomes (L83-100) has carried to the introduction a part of the methods. The choice of genes does not belong here.
L171 what are GTGR and RTGR?
L440 this chapter does not speak about optimisation of the method, and the reader still has no idea about the principle of this method
The discussion has not been changed much and so it remains in part speculative
L863 I do not understand "airways of plant mitochondria". Please revise carefully.
Author Response
Thank you very much for your comments. We took your comments into consideration and made revision. The list of changes and detailed response to all questions are present below.
1. The paper has been improved. Some language problems still remain, for instance in Figure 5,6,7 legend. More work is needed on this.
Thank you for this remark. We fixed legends in these figures.
2. The movement to introduction of the part on mitogenomes (L83-100) has carried to the introduction a part of the methods. The choice of genes does not belong here.
We removed the choice of genes from introduction.
3. L171 what are GTGR and RTGR?
We clarified that R is A or G. A more detailed feature of these sequences was discussed in the Discussion section.
4. L440 this chapter does not speak about optimisation of the method, and the reader still has no idea about the principle of this method
We changed the title of this chapter and added a short discussion of the principle of the method and its advantages.
5. The discussion has not been changed much and so it remains in part speculative
Thank you very much for this remark. We agree with you, that results of current research did not allow to conclude that alternative respiratory pathways in plant mitochondria mediate their resistance to pesticide exposure. But our current results partially confirm the results of our recent work [ref 31 and 36]. We discussed the results of the current study more carefully in the context of our previous researches. More than that, we moderated some conclusions given in the discussion, since our study does not definitively prove the theory, it only provides additional arguments in its favor. We hope the discussion has become less speculative.
6. L863 I do not understand "airways of plant mitochondria". Please revise carefully.
Thank you for this remark. We replaced this phrase by “alternative respiratory pathways of plant mitochondria”